# Analysis of the Face Mask Use by Public Transport Passengers and Workers during the COVID-19 Pandemic

**DOI:** 10.3390/ijerph192114285

**Published:** 2022-11-01

**Authors:** Ekaterina A. Shashina, Ekaterina A. Sannikova, Denis V. Shcherbakov, Yury V. Zhernov, Valentina V. Makarova, Tatiana S. Isiutina-Fedotkova, Nadezhda N. Zabroda, Elena V. Belova, Nina A. Ermakova, Tatiana M. Khodykina, Anton Yu. Skopin, Vitaly A. Sukhov, Anna A. Klimova, Tamara Nikolic Turnic, Irina I. Yakushina, Olga A. Manerova, Vladimir A. Reshetnikov, Oleg V. Mitrokhin

**Affiliations:** 1Department of General Hygiene, Sechenov First Moscow State Medical University (Sechenov University), 119991 Moscow, Russia; 2National Medical Research Center for Urology, Sechenov First Moscow State Medical University (Sechenov University), 119991 Moscow, Russia; 3Department of Chemistry, Lomonosov Moscow State University, 119991 Moscow, Russia; 4F.F. Erisman Federal Scientific Center of Hygiene of Federal Service for Surveillance on Consumer Rights Protection and Human Wellbeing, 141014 Moscow, Russia; 5Department of Pharmacy, Faculty of Medical Sciences, University of Kragujevac, 34000 Kragujevac, Serbia; 6N.A. Semashko Department of Public Health and Healthcare, Sechenov First Moscow State Medical University (Sechenov University), 119991 Moscow, Russia

**Keywords:** face mask, social responsibility, transport workers, passengers, motivational attitudes, adverse reactions to mask wearing

## Abstract

(1) Background: The use of face masks and gloves in public places directly shows the commitment of the population to the established regulations. Public transport is one of the most-at-risk places of contamination. The aim of the study was to analyze the face mask use by public transport passengers and workers during the COVID-19 pandemic. (2) Methods: Public transport passengers and workers were surveyed. Periodic intermittent selective observation was used to gauge the level of adherence to the established regulations among public transport passengers. Factor analysis was used to identify factors determining the face-mask-wearing comfort. (3) Results: The majority of passengers (87.5%) and all transport workers (100%) used face masks and gloves. Most of the users wore only face masks. Only 41.6% of passengers and 74.7% of transport workers wore face masks correctly. Motivational attitudes at the implementation of preventive measures were determined: established regulations in the public place (55.8%) and the protection of one’s own health and the health of family members (44.2%). Only 22.5% of those wearing face masks believed that doing so will have any effect on the spread of an infectious disease, and 10.8% wore masks to maintain the health of people around themselves. A low level of social responsibility was demonstrated. For 53.4% of workers, face mask wearing was uncomfortable. The majority of workers had adverse reactions to mask wearing: feeling short of breath (52.8%), hyperemia of face skin (33.8%), and facial hyperhidrosis (67.4%). (4) Conclusions: The comfort of wearing a mask is determined by adverse reactions occurrence, the properties of the mask, working conditions, and the duration of wearing the face mask. It is necessary to develop recommendations to reduce wearing discomfort. These recommendations, along with methods of raising the social responsibility of the population, can contribute to a greater commitment of the population to non-specific prevention measures.

## 1. Introduction

Human behavior in relation to preventive measures is crucial when infectious disease transmission rates are high. These measures include maintaining self-isolation and social distance, using personal equipment to reduce the infection spread (face masks and gloves), and good hygiene practice [1,2]. The fight against outbreaks of infections goes beyond clinical and biomedical approaches and thus contributes to the development and implementation of preventive measures to preserve public health. To decide the extent of the adoption of preventative measures, it is critical to comprehend how the population reacts to those actions. Public involvement in preventive measures is viewed as a social responsibility in the literature [3].

In order to determine the extent to which preventive measures are being taken, it is important to understand how the population is responding to these actions. For residents of Asian countries, wearing face masks is a common occurrence and was used long before the COVID-19 pandemic [4]. For example, in Japan, wearing face masks has become embedded in everyday practice [5]. In Hanoi, all public transport passengers wore face masks during the current pandemic [6]. Previous studies have identified populations that are more likely to wear face masks. These include people who consider themselves vulnerable to infection with the virus and those who feel responsible for both their health and the health of their loved ones, elderly men and women [7,8], middle-aged women [8,9], urban residents [9], and people who had cases of infection in the family [10]. Face masks are worn less frequently by those who perceive themselves to be immune to the SARS-CoV-2 virus—young people of both sexes [8] and men aged 18 to 39 years [7,8,11]. People wear a face mask more often on public transport than at work or leisure [11] or in other public places [12], and more often in the subway than on above-ground public transport [8].

The mandatory use of face masks and gloves in public places is a preventive measure, the implementation of which directly showing the commitment of the population to the established regulations. The probable reason for why the population unequally adheres to preventive recommendations is an insufficient understanding and use of informational and social impact [13]. The possible reasons for the refusal of face masks may be (1) a low awareness of the population regarding the effectiveness of using a face mask as an individual to reduce the risk of the infection spreading [14,15,16]; (2) a low understanding of the individual about their role in the spread of an infectious disease [10]; (3) a low level of social responsibility [17] because a large number of people consider the mask as a means of their personal protection only [18]. The refusal of wearing face masks and gloves can often be associated with discomfort [16,19,20] and adverse reactions to wearing them, including shortness of breath, claustrophobia [21], a decrease in cardiopulmonary exercise capacity [22], and skin reactions [23,24]. This fact has a negative impact on the fight against this infection and increases the rate of its spread [25].

In Russia, during the pandemic, wearing face masks in public places was mandatory [26]. Control measures were taken to control the wearing of masks and gloves in public transport: social audio and video advertising, visual agitation tools, and memos were placed in metro lobbies, stops, and public transport salons, explaining the meaning and rules for using face mask and gloves; persons without personal equipment were subject to fines and were not allowed to enter public transport. Strict control measures were applied to transport workers: video surveillance during the work shift and fines in case of a violation of the regime [27].

Public transport passengers [28,29] and workers [30,31] are the most-at-risk populations of contamination due to: (1) a high passenger flow density; (2) forced contact with a large number of often infected people who arrive from regions where mass vaccination has not yet been carried out; (3) the mixing of the populations of different districts of the city and different cities and countries; (4) difficulty in maintaining social distance; (5) a low social responsibility of people with symptoms of the disease using public transport. Thus, according to Heinzerling A. et al., the incidence of COVID-19 among workers in the transport industry was 5.2 times higher in the bus and urban transport industries and 3.6 times higher in the air transport industry than in all California industries combined [32]. Therefore, wearing personal equipment that reduces the risk of spreading COVID-19 on public transport by both passengers and workers is especially important. Considering that, in a large metropolis, such as Moscow, a person spends a significant amount of their time in close contact with society in the subway, an analysis of the factors influencing the population’s adherence to preventive measures will reduce the rate of the spread of an infectious disease by formulating a policy to increase the level of social responsibility.

An objective assessment of the attitude of passengers and public transport workers to the use of face masks will form an opinion on the level of social responsibility of the population. This is important for a further study of this phenomenon and to determine the factors that may influence the wearing of medical masks. This, in turn, will provide an opportunity to influence the attitude of the population toward preventive measures.

Studying the adherence to and motivations for wearing face masks will help to develop strategies to increase the culture of face mask wearing and the social responsibility for users’ own health and the health of others. Failure to wear a face mask on public transport increases the risk of infection. This, in turn, increases the burden on the healthcare system on the one hand, and, on the other hand, reduces the use of public transport by residents due to perceived health risks, which entails a loss of economic benefits [33,34].

The aim of the study was to analyze the face mask use by public transport passengers and workers during the COVID-19 pandemic.

## 2. Materials and Methods

A survey method was used. For the purpose of conducting surveys of the Moscow population and the transport workers, Sechenov University employees created questionnaires. The population’s questionnaire asked about the respondents’ demographics, whether they wore masks in public, whether they followed the rules for doing so, whether they were offered incentives to accept or reject preventive measures to fight COVID-19, and how comfortable they felt wearing them. The questionnaire for transport workers included the following blocks of questions: demographic characteristics, working conditions, type and mode of wearing a face mask, and subjective assessment of reactions to wearing a face mask and wearing comfort. Since face-mask-wearing reactions can be linked to heat production and heat transmission during a job, we assessed such working conditions as the primary work place (outdoors or indoors), the availability of an air conditioner, and energy consumption during work. In order to assess the frequency of occurrence of reactions, a scale from 0 to 5 points was used: 0 points if the reaction never occurs, 5 points if it always occurs. The severity of reactions was assessed on the following scale: 0 points for no reaction; 5 points for the most pronounced reaction. In order to assess the wearing comfort, a scale from 1 to 5 points was used: 1 point for discomfort, 5 points for maximum comfort. The anonymous survey was carried out from September–October 2021. Approximately 39% of the population and at least 60% of transport workers in Moscow were vaccinated at the beginning of October 2021 (available online: https://gogov.ru/covid-v-stats/msk (accessed on 23 September 2022)). Transport workers were required to be vaccinated under Decrees of the Chief Public Health Officers in Moscow (No. 1 dated 15 June 2021) [35].

The validation of the questionnaires was carried out in several stages. (1) Development of a questionnaire. A systematic review of the literature was aimed at identifying the types of adverse reactions that may occur when wearing a mask, and the motivation behind and reasons for doing something by people. The formulation of questions was based on clarity and accessibility of perception by respondents, with the exception of the use of specific medical terminology and abbreviations. (2) External validation was carried out using a peer review method (8 employees of the General Hygiene Department and Department of Public Health and Healthcare), which corrected the wording of a number of questions and answers of the questionnaire. All experts were professionals and have been engaged in scientific and pedagogical activities for over 15 years. The average age of the experts was 43 ± 2.1 years. The expert group assessed the correspondence of each question to the conceptual structure. Relevance, clarity, simplicity, and ambiguity were assessed for measuring content validity. (3) The content reliability index was calculated for assessing the relevance, clarity, and simplicity of the elements and the overall reliability of the content of the questionnaires. Its value was 0.82. (4) The study was approved by the local ethical committee of Sechenov University.

We used the recommended methods for calculating the required sample size for the cross-sectional descriptive studies. For the survey, we used 400 people and added a few percent for a non-response, so the final sample size was 470 people [36]. For direct observation, we estimated the simple size, taking into account the following parameters: the increased level of accuracy, a significance level of 0.05, 95% confidence level, and 5% margin of error. The expected proportion in the population was 50%. Thus, the required sample size was 384 people [37]. The factor analysis was carried out for the general database of transport workers (4849 people). The sample size was calculated using the Altman nomogram [38] and was corrected for comparison of independent groups of different sizes (1875 people). A quota selection of respondents by sex and age was made in accordance with the population of Moscow in order to provide a representative sample. In this case, the data presented on the website of the Federal State Statistics Service for Moscow and the Moscow Region [39] as of 1 January 2022 were used; 470 people were chosen at random to make up the sample.

The assessment of the objective level of adherence of public transport passengers to the established preventive measures was carried out by the direct observation method. The observers registered passengers wearing face masks and without them in the Moscow metro coach, and also recorded those who used their mask incorrectly (either a face mask covering only their mouth or a face mask lowered under the chin). The observers remained inconspicuous and avoided interaction with the study object. The data were standardized to the form and time of data collection. The tool for collecting information was the observation card developed by us. The sample consisted of 384 individuals who were chosen at random from the Moscow metro passengers.

A representative sample was chosen from the database of responses from transport workers (4849 persons), matching the sample of Moscow transport users by age and gender characteristics (470 people in total).

Statistical processing of the surveys’ results was carried out using the IBM SPSS Statistics 22 package. The Jarque–Bera test was used to determine whether the distribution was normal. The null hypothesis (H0) that the residuals of the considered indicators of the groups of participants have a normal distribution was accepted by us at the significance level of *p* > 0.05.

The frequencies and percentages were computed for categorical variables. For frequencies, a 95% confidence interval was calculated using the Wilson method. Significance of the differences in presented features was determined using the chi-square test. The null hypothesis (H0) was that the true difference between these group means is absent. Factor analysis using the principal components analysis (normal varimax, at a level >0.600) was applied to those survey questions that demonstrated the largest number of statistically significant indicators of Spearman’s correlation coefficient. The cumulative percentage was 73.47%. When testing statistical hypotheses, a *p*-value of <0.05 was used.

## 3. Results

The characteristics of the study populations in public transport passengers and workers are presented in Table 1.

The increased promotion of personal equipment reducing the risk of spreading COVID-19, and the designation of their importance in maintaining one’s own health and the health of people around the user should have formed a certain way of thinking that determines the role of the individual in the spread of an infectious disease: the use of face masks and gloves reduces the risk of infection, and also minimizes the possibility of an unintentional transmission of the virus. The results of the analysis of the face mask and glove use during the COVID-19 pandemic by public transport passengers and workers are presented in Table 2 and Table 3.

The survey revealed that the majority of surveyed passengers (87.5% (84.0–90.2)) and all transport workers (100%) used personal equipment that reduced the risk of spreading COVID-19. Most of the users wore only face masks.

At the same time, the percentage of passengers ignoring this preventive measure was noted (12.5% (9.8–15.9)). The stated attitude about wearing gloves and face masks in public spaces demonstrates a poor level of social responsibility among the populace, which is brought on by a deliberate disregard for one’s own health and the health of others. Contrary to passengers, the employer used continuous video surveillance during the work shift to monitor whether transport workers were wearing personal equipment that reduced the risk of spreading COVID-19 and imposed fines in the event of noncompliance.

According to survey results, only 74.7% (70.4–78.5) of transport workers and only 41.6% (36.7–46.7) of passengers wore face masks correctly, covering the mouth and nose and fitting tightly on the sides of the face.

Numerous factors affect how often people use face mask and gloves. If these factors are identified, we will be able to discover the factors that drive this behavior. It will be possible to affect the degree of social responsibility based on the information gathered. The following distribution of factors that influence the use of face masks in public transport was discovered through a survey of passengers (Table 4).

Regulations requiring their use in public settings were the main drivers for passengers to wear face masks. The protection of one’s own health and the health of family members was a significant additional factor. A low level of social responsibility in the group of respondents wearing face masks is shown by the tiny percentage of respondents who chose the answer “Maintaining the health of people around you”. At the same time, the sense of a personal role in the fight against infection influences the commitment to compliance with preventative measures, among other factors. However, just 26.6% of those wearing face masks believe that doing so will have any effect on the spread of an infectious disease. Therefore, responders minimize the significance of their behavior, including efforts to combat the infection, which affects the development of personal responsibility to society. According to the survey’s findings, 53.4% of workers found that wearing a face mask was uncomfortable (1–2 points). Rubber bands behind the ears that rub, restricted sight fields, fogged glasses, the sensation that something is present on the face, etc., are all signs of discomfort (Table 5). The majority of respondents, however, remarked that wearing a mask results in feeling short of breath and the occurrence of skin reactions, which show up as hyperemia, peeling, and itching on the face skin after wearing; in addition, there were also facial hyperhidrosis and acne.

According to the survey of transport workers, factors were identified that, to a greater extent, determine the feeling of discomfort toward wearing a face mask.

According to Spearman’s correlation analysis, 15 of the survey’s 30 variables had the most statistically significant correlation coefficient indications. The use of factor analysis allowed for the identification of four factors that identified the face-mask-wearing discomfort (Table 6).

Factors 1 and 2 can be identified as the “Influence of adverse reactions on face-mask-wearing comfort during the work shift”.

Factor 1 is the most informative (27.08%). Its composition is determined by the variables associated with the occurrence of such reactions, which are more related to the chemical composition of the material, microbial contamination of the inner surface of the face mask, and the presence of problematic skin and chronic skin diseases that can be aggravated when wearing a mask.

Factor 2 has a 23.55% informative value. Such reactions, which are more concerned with the mask’s breathability, dictate its composition. We have previously conducted studies on the impact of face mask characteristics on the occurrence of adverse reactions [40,41,42].

Factor 3 covers the variables gleaned from responses to a block of questions about the respondents’ working conditions, and it has an informative value of 15.80 percent. This factor can be identified as the “Influence of working conditions on face-mask-wearing comfort during a work shift”.

Factor 4 includes only one variable from the block of questions “Type of face mask and the wearing mode of it”, with an informative value of 7.04 percent. This block of questions covered the kind of mask material, how many layers it had, how long it was worn for, and the method of face mask wearing. Only the duration of the continued face mask wearing demonstrated the highest significance for wearing comfort from all other indicators.

Thus, the comfort of wearing a mask is influenced primarily by adverse reactions, the occurrence of which, in turn, is influenced by the properties of the mask, then, by working conditions, and, finally, by the duration of wearing a face mask. Adverse reactions may cause a mask to be worn incorrectly, such as on the chin or covering only the mouth. This aspect cancels the mask’s ability to lower the risk of respiratory infections. Additionally, adverse reactions can decrease the attention and concentration of the worker. This, in turn, can potentially cause trauma or/and a lower work productivity. The highlighted factors should be taken into consideration when formulating suggestions to lessen the frequency and severity of adverse responses.

## 4. Discussion

In this study, we analyzed the adherence to wearing face masks of both passengers and transport workers. To determine the correctness of wearing a face mask, we used an observational method instead of a survey. We used factor analysis to determine which factors contributed to the face-mask-wearing discomfort. All of them are the strengths of our work. The limitation of the study is that respondents were not asked about their vaccination status at the time of the survey, since vaccination can give rise to a false sense of security and can be an anti-motivator for wearing face masks, despite the fact that, during vaccination, it is explained that the vaccine protects against the complications of the disease but not against infection and the possibility of transmitting the virus to others while being sick in a mild form. In addition, the limitation is the use of only electronic forms of the questionnaire because only people using the Internet were able to take part in the survey, i.e., our sample was limited by social status.

The results obtained by us are consistent with the results of studies conducted in other countries and in other cities in Russia. Thus, in the Voronezh region of Russia, the percentage of people wearing face masks on public transport was 63.81%, and wearing face masks correctly was 34.24% [43], versus 70.8% and 41.6%, respectively, obtained in our study. The proper way to wear a face mask assumes that it fits snugly to the face; the lower edge of the mask covers the chin, whereas the upper edge covers the tip of the nose and nostrils. In Saudi Arabia, 73% of residents wore face masks during the COVID-19 pandemic on public transport [12], and, in the UK, 82.6% [11]. However, in Hanoi, Vietnam, there is a greater adherence of the population to this measure of nonpharmaceutical prevention: 100% of passengers wore face masks, and only 11% wore them incorrectly [6]. In Ghana, 92.4% of public transport drivers wore masks [7]. However, there was a low adherence of the passengers to the observance of the “face mask regime” on public transport. According to the observation results, the “face mask regime” was observed in only 12.6% of city buses (less than three people were without face masks); in 21.3% of buses, the mask regime was ignored (less than three people wore face masks) [29]. Only 14.32% of surveyed public transport drivers in Gondar, Ethiopia reported that they always wear a mask correctly at work while driving [10].

The population’s proactive commitment to the use of personal equipment during the COVID-19 pandemic surely slows the spread of the virus [44]. According to the survey’s results, face masks were the most often used form of protection. However, it must be kept in mind that only the proper use of a face mask helps to preserve one’s own health and the health of people around them.

The outcomes of periodic intermittent selective observation were examined in this regard. The findings indicated that more than half of the passengers under observation wore their face masks improperly, either by lowering them under their chins (29.0% (24.4–33.7)) or not covering all of their airways (29.4% (24.9–34.3)). These respondents have likely worn face masks as a necessary accessory in a public place, rather than knowingly. Various factors may contribute to this kind of behavior, including: it was difficult to breathe through the nose with a face mask completely covering the upper airways; the lack of a face mask served as a risk of administrative punishment, whereas nothing was expected for its improper use; it was uncomfortable to feel condemning attitudes of society. It should be noted that the study was conducted at the time of the rise in the incidence of COVID-19. This indicates that the general public was informed about the rise in incidence and was required to be aware of the dangers of infection. Because they are aware that there is a high risk of infection in the event that preventive measures are not followed, this group of people’s activities is primarily motivated by the need to follow instructions rather than by the desire to maintain their own and others’ health. This attitude demonstrates an insufficient level of social responsibility, as a result of which, with weak supervision, the number of people who ignore the use of individual equipment that reduce the risk of spreading infection will only grow.

A strict control over the wearing of masks by transport workers contributes to a greater adherence to their use. However, despite video monitoring, only three quarters workers correctly wore a mask covering their mouth and nose. It is likely that the wearing discomfort is the reason for wearing masks incorrectly. According to the factor analysis results, discomfort when wearing a face mask is possibly due to the presence of adverse reactions that are associated with the properties of the face mask, which: depend on the mask material; are due to the working conditions that affect thermal comfort (energy consumption during work, the microclimate at the workplace); are due to the duration of wearing.

The motives identified by us are mainly those identified in other studies; however, their percentage distribution is different. Agyemang E. et al. found that the motive “I do so for my own safety and those of my loved ones” was in first place (94% of respondents chose it). The similar motive “Maintaining both your and your family members’ health” in our study was chosen by only 53% of responders. The motive “impossible to enter public space” was noted by about the same percentage of respondents in that study as in ours (65% and 66.6%, respectively) [7]. The motive “Fear of getting fined”, which ranks third in our study (40% of respondents), is also considered by other authors as significant [7,8]. In studies, other motives for face mask wearing were found: the presence of chronic respiratory diseases [12], knowledge about the ways that the virus spreads and the consequences of infection, as well as knowledge about the effectiveness of respiratory protective equipment [7,10,15,45], examples of other society members [7,8], and reminders from employers and relatives to wear face masks [7].

In Russia, the organization of non-pharmacological preventive measures in the fight against a new coronavirus infection was formed on the policy of punishment, and information was actively disseminated about the introduction of administrative responsibility for refusing to use personal protective equipment in public places. In this regard, the results of the study indicate that, among the proportion of the population who used face masks, one of the most common reasons for wearing them is the fear of receiving this punishment. At the same time, an assessment of British respondents’ preventive behavior found that its driving force is anxiety: those who were more anxious were more likely to take preventive measures, including wearing face masks [46]. A survey of public opinion in the US and Canada indicated that people who object to wearing face masks are a small but very active minority. The media’s coverage of this part of the population may have given the false impression that anti-mask sentiment is widespread. However, our data, as well as public opinion data [16], show that most people are willing to follow the advice of health authorities regarding the wearing of masks.

Face mask use can reduce the rise in incidence [47] but is only beneficial when shared by a substantial percentage of the community [48,49] as opposed to respirators, which offer a tighter fit and more effective filtering of small virus-containing particles [50,51,52]. Face masks are more likely to be attributed to the means of collective rather than individual protection. Many studies have repeatedly indicated the occurrence of adverse reactions to wearing masks [19,23,53,54,55,56], which either leads to the refusal to use them or to incorrect use. Face masks lose their effectiveness when worn improperly. Therefore, it is necessary to develop recommendations to reduce discomfort when wearing face masks and the choice of the face mask type considering the individual characteristics of the user and the conditions of their work. These recommendations, along with methods of raising the social responsibility of the population, can contribute to a greater commitment of the population to non-specific prevention measures.

Measures used to increase passenger adherence to wearing masks can be based on the motivation of those groups who wear masks less frequently—men and young people—raising awareness of mask effectiveness and the perception of the pandemic risk [14,57]. Bogdan I.V. et al. propose the introduction of facilitators reminding them of the importance of wearing face masks in transport. At the same time, the wearing of face masks in the correct state should be positioned as a social norm. It is necessary to strengthen control over the correct wearing of face masks. The personal example of facilitators and controlling people is very important here. All measures should be aimed at strengthening the subjective perception of the importance of measures for one’s own health and the health of others, and not at the fear of a fine [8]. When conducting motivational campaigns, it is important to use all crisis communication strategies. It is extremely important to disseminate timely updated information and to use not only traditional media, journalistic, and broadcasting platforms, but also the practice of searching and exchanging information among the population’s social networks, Twitter user accounts of scientific organizations, and individuals to form dialogues with the population [58]. Sankari S. et al. propose a new automatic notification system through speakers located inside the public transport cabin in case of non-compliance with the rules of wearing face masks or social distancing [59]. Undoubtedly, only the integrated use of all motivational measures can increase social responsibility and the adherence of the population to nonpharmaceutical measures against the SARS-CoV-2 virus and other biological threats in the future.

## 5. Conclusions

A low level of social responsibility and individual understanding of one’s part in the fight against an infectious disease were shown to be contributing factors to the population’s low commitment to taking preventive measures.

The motivational attitudes that affect human behavior in an epidemic (pandemic) for face mask users in Russia were clarified and supplemented, which will make it possible to strengthen measures stimulating the wearing of face masks and will make them more productive.

Approximately half of those who wear a face mask experience discomfort. Face-mask-wearing discomfort is mainly determined by working conditions, mask characteristics, wearing duration, and the presence of adverse reactions to wearing a mask.

It is necessary to develop recommendations to reduce discomfort when wearing face masks, and the choice of face mask should take into account the individual characteristics of the user and the conditions of his work. These recommendations, along with methods of raising the social responsibility of the population, can contribute to a greater commitment of the population to non-specific prevention measures.

## Figures and Tables

**Table 1 ijerph-19-14285-t001:** Characteristics of the study population in public transport passengers and workers.

Demographic Characteristics	Transport Passengers, n (%)	Transport Workers, n (%)
Sex of the caregivers		
Male	227 (48.3)	228 (51.5)
female	243 (51.7)	242 (48.5)
Age of the caregivers		
20–29 years	71 (15.0)	71 (15.1)
30–39 years	149 (31.6)	149 (31.7)
40–49 years	137 (29.2)	137 (29.2)
50–59 years	113 (24.2)	113 (24.0)

**Table 2 ijerph-19-14285-t002:** Use of face mask and gloves.

Variables	Transport Passengers, n (%)	Transport Workers, n (%)	Significance of the Differences
Use face mask and gloves	78 (16.7)	151 (32.1)	χ^2^ = 31.19, *p* < 0.001
Use face mask only	333 (70.8)	319 (67.9)	χ^2^ = 0.98, *p* = 0.322
Do not use anything	59 (12.5)	0	

**Table 3 ijerph-19-14285-t003:** Methods of face mask wearing.

Variables	Transport Passengers, n (%)	Transport Workers, n (%)	Significance of the Differences
Mask covers the nose, mouth, and chin	160 (41.6)	351 (74.7)	χ^2^ = 161.25, *p* < 0.001
Mask covers only the mouth	113 (29.4)	57 (12.1)	χ^2^ = 22.87, *p* < 0.001
Mask is often shifted under the chin	111 (29.0)	62 (13.2)	χ^2^ = 17.20, *p* < 0.001

**Table 4 ijerph-19-14285-t004:** Motivating factors for public transport passengers to wear a face mask.

Motivating Factor	Frequency		95% Confidence Interval
	n	%	
The prerequisite established by the government, without which, it is impossible to enter a public space	313	66.6	62.1–70.8
Maintaining both your and your family members’ health	251	53.4	48.8–57.8
Fear of getting fined	188	40.0	37.6–42.4
Preventing the spread of COVID-19	125	26.6	22.6–30.6
Maintaining the health of people around you	63	13.4	10.3–16.4
Others may judge you for not wearing a face mask	31	6.6	3.2–9.8

**Table 5 ijerph-19-14285-t005:** A subjective assessment of adverse reactions to and discomfort toward wearing a face mask.

Reactions to Wearing a Face Mask	Frequency		95% Confidence Interval
	n ^1^	%	
Discomfort	251 ^2^	53.4	48.8–57.9
Feeling short of breath	248	52.8	48.1–57.3
Headache	109	23.2	19.5–27.3
Hyperemia/peeling/itching of face skin	159	33.8	29.6–38.3
Facial hyperhidrosis	317	67.4	62.9–71.6
Acne	84	17.9	14.6–21.7

^1^ Number of responders who assess frequency of the reaction from 3 to 5 points according to assessment scale (often, very often, always). ^2^ Number of responders who assess wearing comfort from 1 to 2 points according to assessment scale (discomfort, low comfort).

**Table 6 ijerph-19-14285-t006:** A subjective assessment of adverse reactions to and discomfort toward wearing a face mask.

Variable	Factor 1	Factor 2	Factor 3	Factor 4
	Correlation Coefficient			
The frequency of acne	0.876			
The severity of acne	0.872			
The frequency of sneeze and lacrimation	0.828			
The severity of sneeze and lacrimation	0.854			
The frequency of hyperemia, peeling, itching of the face skin	0.637			
The severity of hyperemia, peeling, itching of the skin of the face	0.670			
The frequency of facial hyperhidrosis		0.861		
The severity of facial hyperhidrosis		0.859		
The frequency of feeling short of breath		0.824		
The severity of feeling short of breath		0.803		
Occupational hazard class			0.848	
Level of energy consumption during job			0.770	
Primary workplace (outdoors, indoors)			0.730	
Air conditioning of the workplace			0.697	
Duration of continuous wearing of a face mask				0.952
**Percentage of Variance**	**27.08**	**23.55**	**15.80**	**7.04**

## Data Availability

The data presented in this study are available on request from the corresponding author.

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
