# Peer review of "Analysis of the Face Mask Use by Public Transport Passengers and Workers during the COVID-19 Pandemic"

_ijerph, 2022, doi:10.3390/ijerph192114285_

Round 1

Reviewer 1 Report

Dear authors,

This is a well-written and well-structured paper and I will propose publication, subject to very minor revisions:

-          Firstly please slightly expand the literature review section that appears as Introduction. What key theoretical issues have you been exploring? Why is it important? Do you seek to make a contribution in the field of transportation and public services? Try to elaborate and be more specific.

-          What do you mean by the sociological method of questioning? Can you provide a citation regarding the term? Do you mean surveys? Please elaborate here.

-          The findings are well-structured but in the Discussion and Conclusion, you need to connect your findings with other literatures, as I stated in the first comment above.

Again, this is a very interesting study and I wish you good luck with it,

Author Response

Manuscript ID ijerph-1966847

Response to Reviewers

Thank you for allowing us to submit a revised draft of the manuscript ‘Analysis of the face masks use by public transport passengers and workers during the COVID-19 pandemic’ for publication in the IJERPH MDPI. We appreciate the time and effort you dedicated to providing feedback on our manuscript and are grateful for the insightful comments and valuable improvements to our paper. We have incorporated all of the major corrections and resubmitted the manuscript. Please see below for a point-by-point response to the reviewer's comments and concerns.

Reviewer 1

We thank for reviewing our article and your comments! We tried to make it better, taking into account the changes made.

Point 1: Firstly, please slightly expand the literature review section that appears as Introduction. What key theoretical issues have you been exploring? Why is it important? Do you seek to make a contribution in the field of transportation and public services? Try to elaborate and be more specific.

Response 1: We have changed the section Introduction. Please, find the changes highlighted in the text.

Points 2: What do you mean by the sociological method of questioning? Can you provide a citation regarding the term? Do you mean surveys? Please elaborate here.

Response 2: It was a typo, we meant “survey”. It is corrected in the manuscript.

Point 3: The findings are well-structured but in the Discussion and Conclusion, you need to connect your findings with other literatures, as I stated in the first comment above.

Response 3: We have changed the sections Discussion and Conclusions. Please, find the changes highlighted in the text.

Reviewer 2 Report

The topic of the article is very timely and interesting. Many authors in different countries are conducting research on the impact of COVID pandemic 19 on different areas of activity, including passenger transport (public transport).The use of face masks in public transport is also part of this research.

I make the following comments on the content:

1. There was no review of the literature, both national and global, relating to studies of passenger behaviour on public transport in the COVID -19 pandemic and adherence to the rules imposed (including the wearing of masks). How have individual governments or local authorities dealt with the mandate to wear masks on public transport?

2 There was no scientific discussion taking into account similar researches in other countries. Are the presented research results similar to or different from those presented by other authors? The discussion should be compared to a review of the literature containing research results on similar issues. What were the strengths of the research presented by the authors? What might have interfered with their validity?

Both parts are essential in the article and need to be supplemented. An international or even national perspective is missing.

3) Conclusions should be related to what has already been studied, what should be improved, what measures should be taken, what proposals for further research should be made on the basis of the presented results, so that the topic of wearing masks is completely covered.

4. Too few literature items were used, which is related to comments 1 and 2.

Author Response

Manuscript ID ijerph-1966847

Response to Reviewers

Thank you for allowing us to submit a revised draft of the manuscript ‘Analysis of the face masks use by public transport passengers and workers during the COVID-19 pandemic’ for publication in the IJERPH MDPI. We appreciate the time and effort you dedicated to providing feedback on our manuscript and are grateful for the insightful comments and valuable improvements to our paper. We have incorporated all of the major corrections and resubmitted the manuscript. Please see below for a point-by-point response to the reviewer's comments and concerns.

Reviewer 2

We thank the reviewer for relevant and important recommendations! We revised our submitted manuscript to satisfy the high-quality standards required by the IJERPH.

Point 1: There was no review of the literature, both national and global, relating to studies of passenger behaviour on public transport in the COVID -19 pandemic and adherence to the rules imposed (including the wearing of masks). How have individual governments or local authorities dealt with the mandate to wear masks on public transport?

Response 1: Our manuscript has been changed. Please, find the changes highlighted in the text.

Point 2: There was no scientific discussion taking into account similar researches in other countries. Are the presented research results similar to or different from those presented by other authors? The discussion should be compared to a review of the literature containing research results on similar issues. What were the strengths of the research presented by the authors? What might have interfered with their validity?

Both parts are essential in the article and need to be supplemented. An international or even national perspective is missing.

Response 2: We have changed the sections Introduction and Discussion. Please, find the changes highlighted in the text.

Points 3: Conclusions should be related to what has already been studied, what should be improved, what measures should be taken, what proposals for further research should be made on the basis of the presented results, so that the topic of wearing masks is completely covered.

Response 3: We have changed the section Conclusions. Please, find the changes highlighted in the text.

Points 4: Too few literature items were used, which is related to comments 1 and 2.

Response 4: 31 new references was added to the list.

Reviewer 3 Report

Reviewers comment

In the study titled ‘Analysis of the face masks use by public transport passengers and workers during the COVID-19 pandemic, the authors Shashina et al. surveyed passengers (n=4700) and transport workers (n=470) on wearing protective wear during the COVID-19 pandemic time September-October 2021 in Russia. The survey mainly aimed at the frequency of wearing masks and/or gloves, the correct way of wearing a mask, the level of comfort, etc. Around 12% of the volunteers were not wearing masks or gloves. 

The sample size is appreciable. Although it was mandatory to wear a mask in public, the number of people wearing masks also depends on how well the rules implement and create awareness. Questions on knowledge of COVID-19 spread in the questionnaire would have been better. The manuscript is well written in general. Because of the vaccination, many countries have lifted the mandatory to wear a mask. Hence discussion part of this article is to be focused on how well one can implement the mandatory mask and gloves during future pandemics. The following suggestions might improve the value of the manuscript.

 Effect of the mask on exercise (PMID: 33849908, PMID: 32632523), quantifying face mask comfort (PMID: 34747682), and Skin reaction to gloves (PMID: 32381129) may be included in the introduction or discussion.

  1. Although the study period is mentioned, the vaccination status of the people in the study may also be included, as it may vary from country to country. 

Author Response

Manuscript ID ijerph-1966847

Response to Reviewers

Thank you for allowing us to submit a revised draft of the manuscript ‘Analysis of the face masks use by public transport passengers and workers during the COVID-19 pandemic’ for publication in the IJERPH MDPI. We appreciate the time and effort you dedicated to providing feedback on our manuscript and are grateful for the insightful comments and valuable improvements to our paper. We have incorporated all of the major corrections and resubmitted the manuscript. Please see below for a point-by-point response to the reviewer's comments and concerns.

Reviewer 3

We thank the reviewer for comments and list of related studies. We tried to make it better, taking into account the changes made.

Point 1: Although it was mandatory to wear a mask in public, the number of people wearing masks also depends on how well the rules implement and create awareness. Questions on knowledge of COVID-19 spread in the questionnaire would have been better.

Response 1: We analyzed these questions in our previous study [1], that is why it was not aim of this article.

  1. O. Mitrokhin, E. Shashina, V. Makarova. Use of face masks by students of the medical university during COVID-2019 pandemic. Published: 11 January 2021 by MDPI in The 3rd International Electronic Conference on Environmental Research and Public Health —Public Health Issues in the Context of the COVID-19 Pandemicsession Infectious Disease Epidemiology. DOI: 10.3390/ECERPH-3-08988.

Point 2: Because of the vaccination, many countries have lifted the mandatory to wear a mask. Hence discussion part of this article is to be focused on how well one can implement the mandatory mask and gloves during future pandemics. 

Response 2: We have changed the section Discussion. Please, find the changes highlighted in the text.

Point 3: The following suggestions might improve the value of the manuscript. Effect of the mask on exercise (PMID: 33849908, PMID: 32632523), quantifying face mask comfort (PMID: 34747682), and Skin reaction to gloves (PMID: 32381129) may be included in the introduction or discussion.

Response 3: We added the suggested articles into references.

Reviewer 4 Report

Thanks so much for giving me the opportunity to review this interesting article about an important topic for public health. The article is imperative as it evaluates the compliance of general population for guidelines and instructions to mitigate spread of infectious diseases such as face mask during COVID-19 pandemic. Reviewing population compliance and evaluating reason of negligence of these measures are important to implement more effective measures for combating infectious diseases. The research methodology is nice, logical, and neatly organized. The manuscript has many strengths in presenting the results and explanation of the finding. Few comments might be of interest to be addressed prior publication:

The abstract should be rewritten into structured abstract with subtitles such as purpose, methods, results, and conclusion

The sample size is too small for such survey with all these variables. How come to select the same sample size for the transport workers and passengers while there is a big gap between both populations. This needs to be more justified

The description of the periodic intermittent selective observation methodology is a little bit confusing. I recommend rewriting this paragraph to be more clear

In the results section; sentence starting “Since a face mask wearing; page 6, line 227; it is better to be included in the methods section”

Professional language revision:

Some sentences need to be completed and/or linked with each other, for example:

-          Sentence started with “the proportion of …” Page 2, line 52

-          “Public transport …” Page 2, line 66

-          The presence of adverse reaction… Page 7, line 273

Author Response

Manuscript ID ijerph-1966847

Response to Reviewers

Thank you for allowing us to submit a revised draft of the manuscript ‘Analysis of the face masks use by public transport passengers and workers during the COVID-19 pandemic’ for publication in the IJERPH MDPI. We appreciate the time and effort you dedicated to providing feedback on our manuscript and are grateful for the insightful comments and valuable improvements to our paper. We have incorporated all of the major corrections and resubmitted the manuscript. Please see below for a point-by-point response to the reviewer's comments and concerns.

Reviewer 4

Thanks for your comments on the manuscript! We tried to make it better, taking into account the changes made.

Point 1: The abstract should be rewritten into structured abstract with subtitles such as purpose, methods, results, and conclusion

Response 1: The abstract has been structured. Please, find the changes highlighted in the text.

Point 2: The sample size is too small for such survey with all these variables. How come to select the same sample size for the transport workers and passengers while there is a big gap between both populations. This needs to be more justified.

Response 2: Due the first part of the manuscript includes the results of the cross-sectional study and we used only descriptive statistics methods (tables 2,3,4,5), we have used the recommended methods for calculation required sample size for such study design. According to different formulas and tables, the required sample size is from 350 to 400. According to various authors, such a sample size provides sufficient power for the study [1,2,3,4]. For survey we used 400 and added some percent for non-response, so the final sample size was 470 [3]. For observation study we estimated the simple size, taking into account the following parameters: the increased level of accuracy, the significance level - 0.05, 95% confidence level, and 5% margin of error were used, expected proportion in population 50%. So, the required sample size was 384 [1].

We deliberately chose samples of the same size, corresponding to each other in terms of sex and age structure, so as not to carry out additional calculations to correct the total study volume, taking into account the analysis of two unrelated samples of different sizes.

The factor analysis presented in Table 6 was carried out for the general database of transport workers (4849 people). The calculation of the sample size was carried out in several stages. Firstly, the frequency distribution of responses was analyzed based on the pilot study. Then, the sample size was calculated using the Altman nomogram [5]. Next, the sample size was corrected for comparison of independent groups of different sizes (it was minimum 1875). And finally, the minimal size of one of the compared groups was calculated using the Lera formula (it was minimum 97) [6].

  1. Charan J, Biswas T. How to calculate sample size for different study designs in medical research? Indian J Psychol Med. 2013 Apr;35(2):121-6.
  2. Otdelnova K. A. Determination of the required number of observations in social and hygienic research. Sat. Proceedings of the 2nd MMI. 1980; 150(6): 18–22.
  3. Paniotto VI, Maksimenko VS Quantitative methods in sociological research. Kyiv: Naukova Dumka. 1982: 272.
  4. Fox N., Hunn A., and Mathers N. Sampling and sample size calculation The NIHR RDS for the East Midlands/Yorkshire & the Humber 2007. Available from: https://www.academia.edu/22574561/
  5. Altman D.G. How large a sample? In: Gore SM, Altman DG (eds.). Statistics in Practice. London, UK: British Medical Association. 1982: 100.
  6. Lehr R. Sixteen s-squared over d-squared: a relation for crude sample size estimates. Statistics in medicine. 1992; (11): 1099-1102.

We have changed the paragraph in the section Methods: “We used the recommended methods for calculating the required sample size for the cross-sectional descriptive studies. For the survey, we used 400 people and added a few percent for non-response, so the final sample size was 470 people [36]. For direct observation, we estimated the simple size, taking into account the following parameters: the increased level of accuracy, the significance level - 0.05, 95% confidence level, and 5% margin of error were used. The expected proportion in the population was 50%. So, the required sample size was 384 people [37]. The factor analysis presented in Table 6 was carried out for the general database of transport workers (4849 people). The sample size was calculated using the Altman nomogram [38] and was corrected for comparison of independent groups of different sizes (it was 1875 people).”

Point 3: The description of the periodic intermittent selective observation methodology is a little bit confusing. I recommend rewriting this paragraph to be more clear.

Response 3: We have rewriting this paragraph: “The assessment of the objective level of adherence of public transport passengers to the established preventive measures was carried out by the direct observation method. The observers registered passengers wearing face masks and without them in the Moscow metro coach, and also recorded those who used their mask incorrectly (either a face mask covered only their mouth or a face mask was lowered under the chin). The observers remained inconspicuous and avoided interaction with the study object. The data were standardized to the form and time of data collection. The tool for collecting information was the observation card developed by us. The sample consisted of 384 individuals who were chosen at random from the Moscow metro passengers”.

Point 4: In the results section; sentence starting “Since a face mask wearing; page 6, line 227; it is better to be included in the methods section”.

Response 4: That was done. We have changed the section Method. Please, find the changes highlighted in the text.

Point 5: Some sentences need to be completed and/or linked with each other, for example: Sentence started with “the proportion of …” Page 2, line 52.

Response 5: We delayed this sentence.

Point 6: Some sentences need to be completed and/or linked with each other, for example: “Public transport …” Page 2, line 66.

Response 6: We have changed this paragraph: “Public transport passengers [28,29] and workers [30,31] are most-at-risk populations of contamination due to: (1) high passenger flow density; (2) forced contact with a large number of often infected people who arrived from regions where mass vaccination has not yet been carried out; (3) mixing of the population of different districts of the city, different cities and countries; (4) difficulty maintaining social distance; (5) low social responsibility of people with symptoms of the disease using public transport.”

Point 7: Some sentences need to be completed and/or linked with each other, for example: The presence of adverse reaction… Page 7, line 273

Response 7: We have changed this paragraph: “According to the factor analysis results, discomfort when wearing face mask is possibly due to the presence of adverse reactions that are associated with the properties of the face mask, which depend on the mask material; due to the working conditions that affect thermal comfort (energy consumption during work, the microclimate at the workplace); and due to the duration of wearing.”

Round 2

Reviewer 2 Report

Dear Authors,

Thank you for considering the suggestions. I don't have any further comments.